# Deployment of a retinal determination gene network drives directed cell migration in the sea urchin embryo

**Megan L Martik[1]\*, David R McClay[2]\***

[1]University Program in Genetics and Genomics, Duke University, Durham, United States; [2]Department of Biology, Duke University, Durham, United States

**Abstract** Gene regulatory networks (GRNs) provide a systems-level orchestration of an organism's genome encoded anatomy. As biological networks are revealed, they continue to answer many questions including knowledge of how GRNs control morphogenetic movements and how GRNs evolve. The migration of the small micromeres to the coelomic pouches in the sea urchin embryo provides an exceptional model for understanding the genomic regulatory control of morphogenesis. An assay using the robust homing potential of these cells reveals a 'coherent feed-forward' transcriptional subcircuit composed of Pax6, Six3, Six1/2, Eya, and Dach1 that is responsible for the directed homing mechanism of these multipotent progenitors. The linkages of that circuit are strikingly similar to a circuit involved in retinal specification in *Drosophila* suggesting that systems-level tasks can be highly conserved even though the tasks drive unrelated processes in different animals.

## Introduction

Much research has been done to understand the cell biology underlying the events of directed cell migration; however, how the events are encoded in the genome and how gene regulatory networks (GRNs) control this process are works in progress. Cell migration is the directed movement of a single cell or a collective group of cells through the early embryo or the adult body. Many different cell types undergo chemotaxis-dependent directed migration during development, including primordial germ cells (PGCs), *Drosophila* border cells, the zebrafish posterior lateral line, *Drosophila* tracheal cells, vertebrate neural crest cells, and vertebrate anterior mesoderm (*Reig et al., 2014*). The ability for cells to undergo directed migration towards a target location requires the use of different signal transduction mechanisms to remodel their actin cytoskeleton in a directed fashion such that they extend filopodia, lamellipodia, and blebs to create a polarized leading edge. In the sea urchin, a near complete developmental GRN describes the specification of endomesoderm (*McClay, 2011*; *Peter and Davidson, 2011*). Studies of this specification network have made the sea urchin a viable model for extending the study of how GRNs can explain control of complex cell behaviors (*Saunders and McClay, 2014*). The migration of the sea urchin small micromeres serves as a powerful experimental model for connecting the genomic regulatory control of morphogenesis to an upstream GRN.

Small micromere cells arise from an asymmetric cleavage of the micromeres at the embryonic fifth cleavage (*Figure 1A*). These cells divide once within the vegetal plate to produce eight cells (*Pehrson and Cohen, 1986*). The eight small micromeres migrate along with the growing archenteron during gastrulation until they reach the animal pole (*Yajima and Wessel, 2012*; *Campanale et al., 2014*). Post-migration, the small micromeres incorporate into the coelomic pouches, which are found on either side of the developing esophagus (*Hyman, 1955*; *Pehrson and Cohen, 1986*; *Luo et al., 2012*).

The coelomic pouches, a mesodermal sub-type, appear at the tip of the forming archenteron. Their specification is initiated early in development by Delta/Notch signaling (*Sherwood and McClay, 1999*;

**\*For correspondence:** megan.
martik@duke.edu (MLM);
dmcclay@duke.edu (DRM)

**Competing interests:** The
authors declare that no
competing interests exist.

**Reviewing editor:** Marianne E
Bronner, California Institute of
Technology, United States

**eLife digest** Within an animal embryo, groups of cells tend to move, or migrate, between different areas before they form into tissues and organs. These cell migrations are regulated by hundreds of genes, which must be expressed at the right time and in the right place. Cells use proteins called transcription factors to regulate the expression of genes. These proteins work together in circuit board-like networks called gene regulatory networks in order to drive different aspects of development, including cell migration.

The sea urchin is a useful model organism to study how animals develop. This is because these marine animals express many of the same genes as humans, but they can be easily manipulated and studied in the laboratory. In a developing sea urchin embryo, cells called the small micromeres move towards one end of animal and get incorporated into a pocket-like structure known as the coelomic pouch. From this pouch, these cells mature and eventually contribute to the adult germ cells (the precursors to the sperm and eggs).

Martik and McClay have now analyzed how small micromeres make their way to their final location in the coelomic pouch. Micromeres were labeled with a dye that fluoresces green so that they could be tracked under a microscope. This revealed that, like other moving cells, micromeres actively change their shape as they migrate. Furthermore, when micromeres were experimentally moved to abnormal locations in the sea urchin embryo, they were still able to actively home in on the coelomic pouch no matter their starting location.

Martik and McClay then identified five transcription factors expressed in the coelomic pouch in the sea urchin that are involved in this homing activity. Reducing the expression of any of these transcription factors was enough to hinder the ability of the micromeres to find their way to the coelomic pouch. Further experiments and analysis then revealed that these five transcription factors work together in a sub-circuit, which is in turn embedded in a larger gene regulatory network.

This sub-circuit that drives cell migration is unexpectedly similar to another circuit in the fruit fly *Drosophila*. Intriguingly, the sub-circuit in the fly controls eye development, which is unrelated to cell homing and migration. These observations raise the possibility that this circuit has been conserved as a unit over millions of years of evolution and redeployed in new networks under completely different circumstances. The data also suggest the possibility that additional conserved sub-circuits will be identified as more systems are analyzed in detail.

*Sweet et al., 2002*). During gastrulation, several other mesodermal cell types undergo epithelial-to-mesenchymal transitions (EMTs) into the blastocoel where they take on different roles in the embryo. The mesodermal cell sheet remaining at the tip of the archenteron at the end of gastrulation forms the two coelomic pouches on either side of the foregut (*Figure 1A*). Only those small micromeres that reach the left coelomic pouch, which will become the future adult rudiment, will survive until adulthood. During metamorphosis of the indirect developing sea urchin, the embryonic small micromeres incorporate into the adult rudiment's left somatocoel, which will later give rise to the gonads of the adult animal and possibly other tissues (*Hyman, 1955*).

Previous research has suggested that the small micromeres contribute to the adult PGCs; however, it has not been shown directly whether the small micromeres contribute to additional adult tissues or whether the only source of the adult PGCs are the small micromeres (*Pehrson and Cohen, 1986*; *Voronina et al., 2008*; *Juliano et al., 2010a*; *Juliano et al., 2010b*; *Yajima and Wessel, 2010*; *Yajima and Wessel, 2012*; *Wessel et al., 2014*). The possibility remains that the small micromeres go on to produce multiple cell types including, but not limited to PGCs (*Yajima and Wessel, 2015*).

Recent publications tracked small micromeres as they moved from the vegetal pole to the coelomic pouches (*Yajima and Wessel, 2012*; *Campanale et al., 2014*). In Yajima et al., it was observed that during gut invagination, the small micromeres did not change position relative to the adjacent mesoderm cells of the advancing archenteron. It was concluded that once they reach the tip of the archenteron, the small micromeres must actively migrate to the left and right coelomic pouches (*Yajima and Wessel, 2012*). Seemingly contradictory evidence from Campanale et al. described an active migration throughout gastrulation and post-gastrulation as they make their way to the coelomic pouch (*Campanale et al., 2014*). While both studies conclude that there is an active

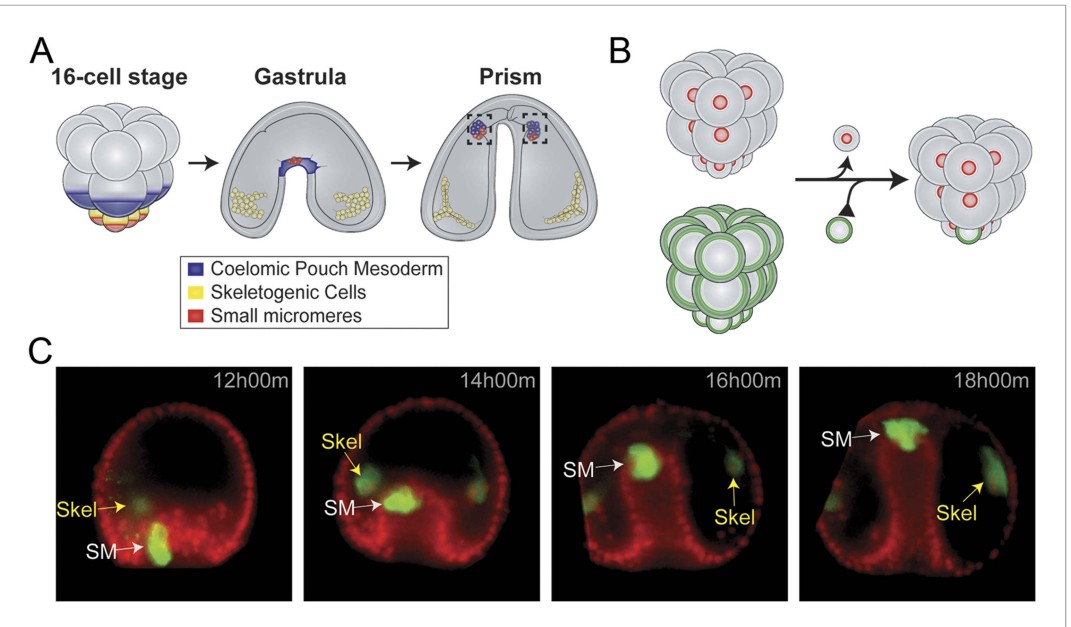

**Figure 1**. Small micromere movements during gastrulation. (**A**) The small micromeres arise at the vegetal pole at the asymmetric fifth cleavage from the micromere lineage (red). During gastrulation, they remain at the tip of the gut. They migrate through the top of the blastocoel and enter the posterior half of the coelomic pouches by prism stage. (**B**) To study small micromere movements and migration at a higher resolution, membrane-GFP-labeled micromeres were transplanted to a H2b-RFP-labeled host. (**C**) Small micromeres actively changed shape throughout gastrulation (extend filopodia and lamellipodia) until they reach the tip of the archenteron. Skeletogenic lineages are labeled as 'Skel', and small micromeres are labeled as 'SM'. See also, *Video 1*.

migration post-gastrulation, we clarified whether the small micromeres acquired their active movement before or after gastrulation. When removed from their endogenous location and placed ectopically, we find that the small micromeres are autonomous and active while migrating throughout gastrulation to make it 'home' to the coelomic pouch. Their active motility during gastrulation is essential for the ectopic small micromeres to undergo directed 'homing' migration to the coelomic pouch.

In order to understand this homing ability at a systems level, we took advantage of the robust homing nature of the small micromeres to determine the GRN at work during their migration. Using a homing assay to quantitatively score the ability of the coelomic pouch cells to direct movement of the small micromeres, we constructed a GRN that governs effector genes directly responsible for the homing mechanism. We find that a specific subcircuit composed of Pax6, Six3, Six1/2, Eya, and Dach1 that is canonically found in retinal development has been deployed for this distinct developmental process. Here, we describe a GRN for homing behavior and uncover a striking similarity of our morphogenesis GRN to the retinal determination gene network (RDGN).

## Results

Underlying GRNs specify each of the coelomic pouch cell types (oral and aboral mesoderm) and the small micromeres early in development (*Oliveri et al., 2006*; *Peter and Davidson, 2011*; *Materna et al., 2012*). We sought to understand the robust nature of the small micromere migration and how that migratory event is transcriptionally controlled. Given the ability to experimentally move the small micromeres to ectopic locations, we first learned that the small micromeres had a remarkable ability to home to the coelomic pouches. Using that observation, we developed an assay to study how the small micromeres underwent a directed cell migration. The assay was also used to score for transcriptional regulation of the migration after perturbing specific transcription factors.

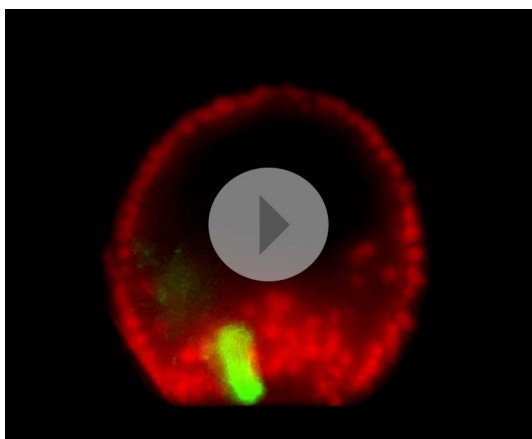

**Video 1.** Movement of small micromeres throughout gastrulation. Membrane-GFP labeled micromeres were transplanted to a H2b-RFP-labeled host. An image was acquired every 3 min for 6 hr starting at 12 hr post fertilization. Videos were projected from multiple z-stacks (every 3 μm, spanning the blastocoel of the embryo) using SoftWorx for DeltaVision. AVIs were then rotated and translated (due to swimming embryos) in FIJI. Frame rate = 5 frames per second.

## Small micromeres actively migrate throughout gastrulation

The mechanisms by which the small micromeres make their way to the coelomic pouch is a topic of debate: one paper cites an active migratory event while another a passive translocation during gastrulation (*Yajima and Wessel, 2012*; *Campanale et al., 2014*). We followed the migratory process at a higher cellular resolution using time-lapse to resolve the seemingly contradictory data on the movement of small micromeres during gastrulation.

Small micromeres were fluorescently labeled, so they could be uniquely followed throughout gastrulation allowing a finer analysis of their behavior than if those cells were not specifically identified (*Figure 1B* and *Video 1*). Host embryos were labeled by injection of histone2B-RFP mRNA and micromere-donor embryos with a membrane-GFP mRNA. At the 16-cell stage, one donor fluorescent micromere was transplanted to the vegetal pole where it replaced a surgically removed micromere (*Figure 1B*). Each fourth cleavage micromere gives rise to two lineages: the small micromere (the proposed primordial germ cell) and the large micromere (the skeletogenic mesoderm). The transplant surgery was performed at the 16-cell stage to ensure the survival of the small micromere daughters for two reasons. First, it assured that the small micromeres (which arise at fifth cleavage by an asymmetric division of the micromere) were not damaged by separation from the large micromeres during micromanipulation since they had not yet been born. Second, when the small micromeres appeared, their only initial connection to the embryo was via the spindle remnant to the large micromere—this attachment to the large micromere was crucial for the incorporation of the transplant into the host embryo. Both reasons for surgically transplanting the micromere ensured the re-establishment and survival of a high percentage of small micromeres in the host embryo (*Table 1* shows scoring of transplant efficacy). At the beginning of gastrulation, this recombinant approach resulted in 8 fluorescent large micromere progeny and 2 fluorescent small micromeres (*Figure 1C* and *Video 1*). It was easy to distinguish the donor small micromere progeny from the large micromere progeny when the large micromeres fused with non-fluorescent host mesoderm cells to form the skeletal syncytium thereby greatly diluting their fluorescence (*Figure 1C* and *Video 1*).

Chimeric embryos made in this manner were examined over six hours of development to follow cell behavior. As seen in *Figure 1C* and *Video 1*, throughout invagination of the archenteron, the small micromeres were highly active. They underwent many cell shape changes, blebbing, and pseudopodia extensions as was also seen by Campanale et al. They extended filopodia and lamellipodia through the basal lamina surrounding the archenteron. Nevertheless, they retained an adhesion and were not displaced relative to the adjacent invaginating epithelial cells (*Figure 1C* and *Video 1*, 12 hr 00 min–18 hr 00 min).

When gastrulation was completed at 20 hpf (early prism stage), small micromere behavior changed in tandem with a change in the basal lamina. At a time corresponding to the end of gastrulation, laminin immunostaining indicated that the basal lamina was undergoing remodeling, as there were large gaps in the laminin component of the basement membrane at the tip of the archenteron (*Figure 2A*). Through those gaps, the small micromeres delaminated (*Figure 2A*), moved to the blastocoelar side of the remodeling basal lamina, and then actively migrated to the posterior end of the left or right coelomic pouch where they were then re-enveloped with a basal lamina layer by 2 dpf (*Figure 2B*). The migratory behavior of the small micromeres suggested that they underwent an EMT at the beginning of their migratory phase over the archenteron and to their coelomic pouch. These cells de-adhered, breached a basal lamina, changed shape, actively migrated within the blastocoel,

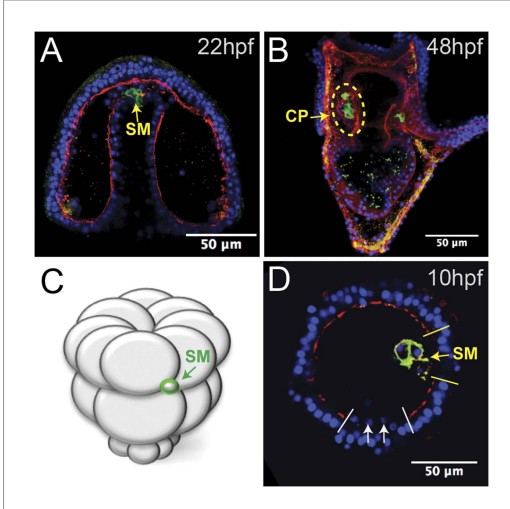

**Figure 2**. Laminin remodeling at the tip of the archenteron facilitates migration of the small micromeres to the posterior end of the coelomic pouch. (**A**) Once the micromeres reach the tip of the gut, they undergo an epithelial–mesenchymal transition (EMT). By immunostaining, we see that laminin (red), at the time, is reduced as the small micromeres (SM, green) breach the top of the gut. (**B**) Once they reach the posterior coelomic pouch, laminin (red) surrounds the small micromeres (green) and NSM to encapsulate the coelomic pouch (yellow dashed circle). (**C**, **D**) Ectopically placed small micromeres underwent an EMT coincident with the endogenous EMT event of the skeletogenic cells. Laminin (red) is absent both at the site of skeletogenic cell ingression (indicated by white arrows at the site of ingression) and at the site of ectopic small micromere (SM, green) ingression (indicated by a yellow arrow).

and then rejoined an epithelium, demonstrating a number of EMT, and perhaps mesenchymal–epithelial transition (MET), properties.

## Small micromeres 'home' to the coelomic pouches from ectopic positions

In many embryonic systems, cells home, or migrate in a directed fashion, from their place of origin to a distant target site. Since that movement is directed, in each case there must be a mechanism to provide guidance (*Richardson and Lehmann, 2010*; *Yajima and Wessel, 2012*; *Campanale et al., 2014*; *Reig et al., 2014*). Knowing that the small micromeres were actively changing cell shape then moving to a new location at the end of gastrulation, we tested to see if small micromeres would undergo a directed migration to the coelomic pouch when placed in an ectopic location. Normally, small micromeres start their trajectory at the vegetal pole, which is the most posterior location in the embryo and terminate in the coelomic pouches near the anterior end of the embryo. Accordingly, small micromeres were placed ectopically at varying locations along the anterior/posterior axis of 16- to 60-cell staged embryos, and the frequency of their ability to find their way to the coelomic pouch at 2 days post fertilization (dpf) was scored (*Figure 3*). Micromeres placed at the vegetal pole served as a control since that is their endogenous starting location. Donors placed in the endogenous location homed 86.6% (n = 213) of the time (*Figure 3A*). 13.4% of observed cases that were considered 'no homing' were instances where we could not find fluorescent small

micromeres in the host embryo's coelomic pouch. Those that did not make it to the coelomic pouch became lost in the blastocoelar space, which we scored as 'no homing'. Micromeres were then placed ectopically between the Veg1 and Veg2 tier and in the animal half between the An1 and An2 tier of presumptive ectodermal cells (scored together as equator), at the animal pole, or in the blastocoel. No matter where they were placed, the vast majority of small micromeres transplanted to ectopic positions anywhere in the embryo made it home (Equator = 93.7% (n = 74), Animal Pole = 78.6% (n = 11), Blastocoel = 78.6% (n = 11)) (*Figure 3B–D*). Interestingly, the majority of transplants were scored to be in the left coelomic pouch after homing suggesting a survival mechanism to ensure the proposed primordial germ cells make it onto adulthood to contribute to the gamete pool.

Next, we wanted to see how the small micromeres homed. Did they move from the ectopic position within the plane of the epithelium, or did they ingress from the ectopic location into the blastocoel and find the coelomic pouch from there? If they did ingress into the blastocoel, were they first attracted to the endogenous small micromeres before migrating to the coelomic pouch mesoderm, or did the small micromeres autonomously move to the coelomic pouches? Could it be that the ectopically placed small micromeres precociously ingressed through the basement membrane and thus headed directly to their final destination? To see when and where the ectopic small micromeres found their way home, we transplanted differentially labeled micromeres to two different locations (one at the animal pole [*Figure 3—figure supplement 1A*] and one at the equator [*Figure 3—figure supplement 1B*]) to host embryos. We then assayed at intervals up to 24 hpf and

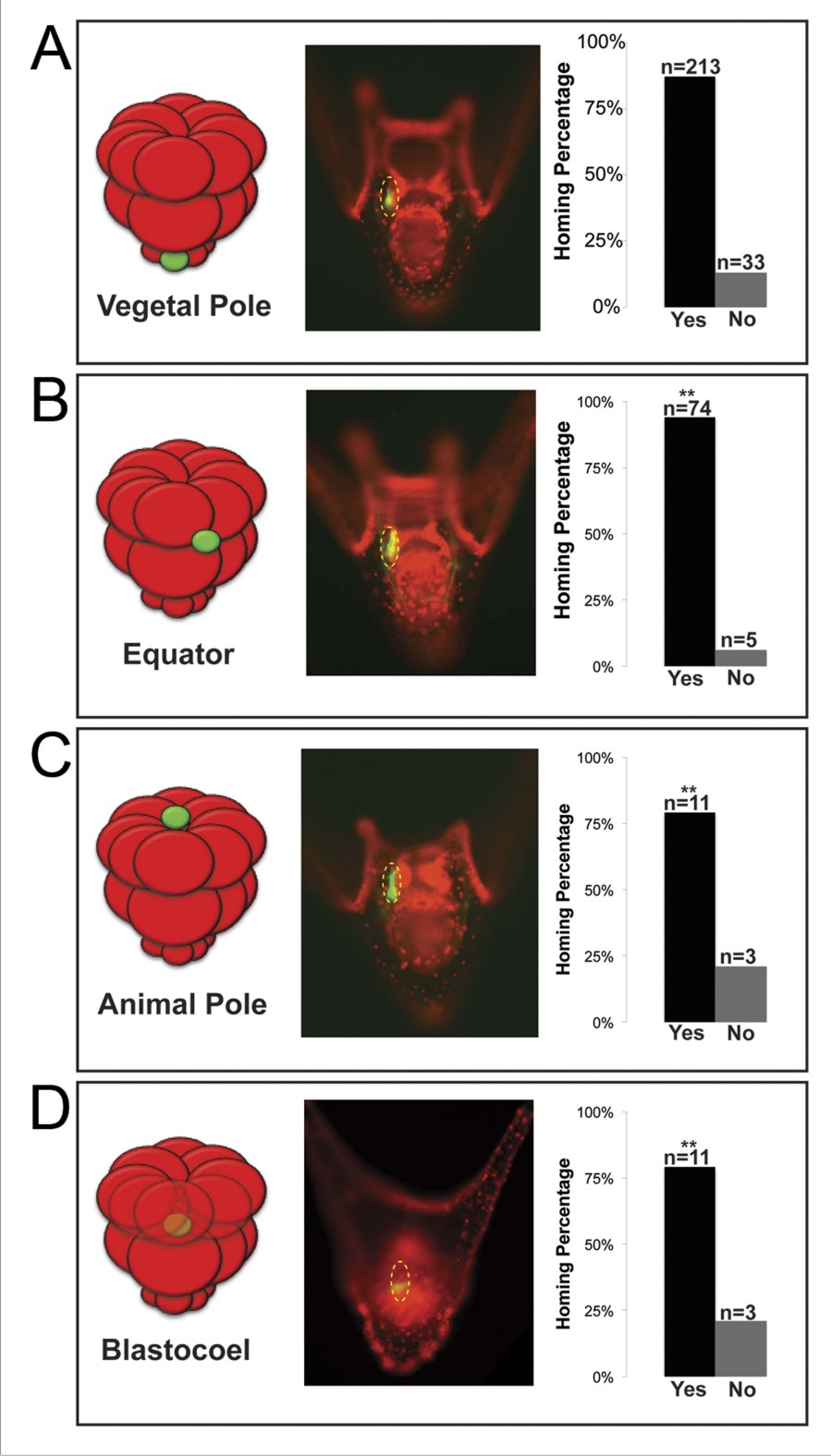

**Figure 3**. Ectopically placed micromeres are able to find their way to the coelomic pouches via a directed homing mechanism from any ectopic location. (**A**) The endogenous, vegetal pole, (**B**) the equator (p < 0.004), (**C**) the animal pole (p < 0.004), or (**D**) inside of the blastocoel (p < 0.004). Illustrations on the left designate their ectopic placement (ectopic micromere = green, host embryo = red). Yellow dashed lines indicate the location of the whole coelomic pouch with the ectopically placed micromeres labeled in green. Graphs depict the percentage of embryos scored

*Figure 3. continued on next page*

*Figure 3. Continued*

with the given ability to home (Yes vs No), 'n' equals the number of embryos scored with the phenotype, and p-values were calculated using a $\chi^2$ test. '**' denotes a statistically significant (p < 0.05) p-value.

The following figure supplement is available for figure 3:

**Figure supplement 1**. Ectopic small micromeres arrive at the coelomic pouches independently.

saw that the ectopically placed micromeres arrived independently of the endogenous micromeres (*Figure 3—figure supplement 1A,B*).

The transplanted small micromeres remained at the ectopic site until the primary mesenchyme cells (PMCs) ingressed. At that time (9–10 hpf), the ectopic small micromeres also ingressed into the blastocoel (*Figure 2C,D*). Because the ectopic small micromeres ingressed at the time of skeletogenic cell ingression, we wondered whether isolated small micromeres could ingress on their own. By ectopically transplanting just 32-cell stage small micromeres (as opposed to the usual 16-cell stage micromere, which would result in large and small micromere cells) (*Figure 2C*), we observed the small micromeres autonomously invade through the laminin layer independent of large micromeres. The ectopic small micromeres breached the laminin layer (*Figure 2D*, yellow arrow) at the same time as the PMC EMT (*Figure 2D*, white arrows) suggesting that the small micromeres create the hole through which they breach and invade, even though the endogenous small micromeres actually breach the basement membrane about 6–8 hr later when they leave the tip of the archenteron.

## Small micromeres home to coelomic pouch precursor cells

Once we learned that the ectopically placed small micromeres home to the coelomic pouch from any location in the embryo, we next investigated whether the signal to which they were responding was coming from the coelomic pouch mesoderm. The question was whether the small micromeres home directly to the coelomic pouch region or indirectly. First, to ask if the mesoderm is the lineage to which the small micromeres home, we knocked out Delta, a signal necessary for mesodermal specification (*Sherwood and McClay, 1999*; *Sweet et al., 2002*). The lack of mesoderm, indeed, had an effect on homing. In the experiment, Delta was knocked down in host embryos and control (unperturbed) micromeres were placed ectopically at the equator of those embryos (*Figure 4A*). The small micromeres did not home to any significant location in the embryo (coelomic pouches were non-existent in the KD) and became lost in the blastocoelar space as is seen in cases we deemed 'no homing' 65% of the time (n = 23, p < 0.0001) (*Figure 4C*). 35% of the time, there was an incomplete knockdown of the Delta signal resulting in a coelomic pouch and the small micromeres homed to that location. Further investigation of the cell types and sub-territories within the coelomic pouch allowed us to elaborate on the genomic control underpinning the homing mechanism.

## Coelomic pouch transcription factors affect small micromeres' ability to home

Next, we wanted to know which transcription factors within each territory of the final coelomic pouch (aboral or oral mesoderm) were upstream of the homing mechanism. We surveyed a list of known coelomic pouch and small micromere transcription factors, and genes found to maintain expression levels during homing were included in our list of potential upstream regulators. The genes further investigated were (1) oral coelomic pouch transcription factors FoxC, FoxF, PitX2; (2) aboral coelomic pouch transcription factors Dach1, Pax6, SoxE, Six3, Six1/2, and transcriptional co-activator Eya; and (3) small micromere transcription factors FoxC and Pitx2. Accession numbers are found in *Table 2*.

The experiment was to transplant a labeled micromere to a differentially labeled host embryo and assay the small micromeres' ability to home in either of the two perturbed scenarios: either with a morpholino-perturbed host embryo or morpholino-perturbed micromere transplant. For each transcription factor, we transplanted control micromeres to control hosts (a positive control for homing), perturbed micromeres transplanted to control hosts, and control micromeres transplanted to perturbed hosts. Transplants were done at the 16-cell stage and scored at two days post-fertilization. Transplant efficacy was also scored and can be found in *Table 1*. Only those transplants

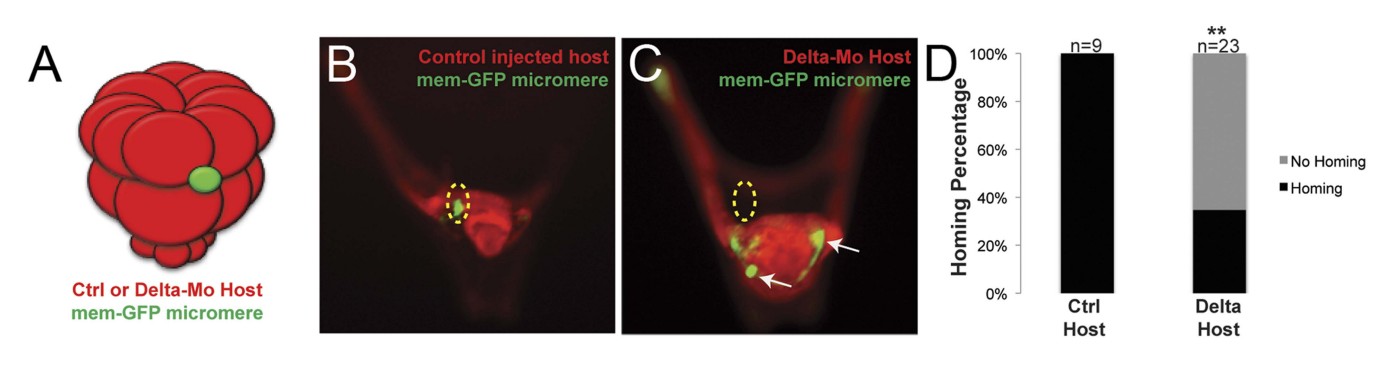

**Figure 4**. Specification of the non-skeletogenic mesoderm (NSM) by Delta signaling is required for homing. Control micromeres labeled with membrane-GFP were ectopically transplanted to either a control TMR (red) host or a Delta–MO TMR (red) host (**A**). When control micromeres were transplanted to a control host, homing of the small micromeres occurs 100% (n = 9) of the time (**B**). When NSM specification was blocked in a host embryo using a delta morpholino, homing of transplanted control micromeres was significantly reduced and only homed 53% of the time (n = 23, p < 0.0001) (**C**, white arrows). (**D**) The graph depicts the percentage of embryos (Control Host vs Delta Host) seen with the given ability to home (Black = Homing vs Gray = No Homing), 'n' equals the number of embryos scored in each case, and p-values were calculated using a $\chi^2$ test. '**' denotes a statistically significant (p < 0.05) p-value.

containing small micromere were scored for homing or no homing. Failed transplants were cases where no small micromeres were observed in the embryo.

One transcription factor, Forkhead family member FoxC, was found to affect small micromere homing when only knocked down in the micromere. FoxC is expressed in both the oral coelomic pouch mesoderm and the small micromeres; however, only when it was perturbed in the small micromere did it affect homing, and only 26% (n = 19, p < 0.001) of the time did the small micromeres successfully make it to the coelomic pouch (**Figure 5A**).

The first coelomic pouch transcription factor we found to be involved in homing was a Ski/Sno family member, Dachshund1. When control micromeres were transplanted to a Dachshund1 morpholino perturbed host, the control small micromeres were only able to home in 10% (n = 10, p < 0.0001) of scored cases (**Figure 5B**). In contrast, when Dach1 was knocked down in the small micromeres only, and the host embryo was a control, homing ability was scored as successful in 77% of cases. We concluded that Dach1 was necessary in the targeted tissue for homing to occur.

Next, we found the paired box family transcription factor Pax6 was upstream of the small micromeres' ability to home. When control micromeres were transplanted to a Pax6 knockdown host, the small micromeres were only able to home 40% (n = 10, p < 0.004) of the time (**Figure 5C**). We then found the six family transcription factor, Six3, to regulate homing. When control micromeres were transplanted to a Six3 knockdown host, the small micromeres were able to home 11% (n = 9, p < 0.00009) of scored cases (**Figure 5D**). Next, we saw transcriptional co-activator and tyrosine phosphatase, Eya, to exhibit an effect on homing. When control micromeres were transplanted to an Eya knockdown host, small micromeres were able to home 28.5% of the time (n = 7, p < 0.002) (**Figure 5E**). Finally, we found another six family transcription factor, Six1/2, to affect homing. Control micromeres transplanted to a Six1/2 knockdown host were able to home 65% (n = 17, p < 0.02) of scored cases (**Figure 5F**). This effect is modest, but statistically significant relative to controls within that experiment. Transcription factors FoxF, SoxE, FoxC, and Pitx2 did not appear necessary for homing when perturbed in the coelomic pouch mesoderm (**Figure 5—figure supplement 1**).

It is important to note that when the same knockdowns are stained for endogenous *vasa* expression (labeling the small micromeres), small micromeres are still able to reach the coelomic pouches (**Figure 5—figure supplement 2**). We observe that the control endogenous small micromeres are within the coelomic pouches in tight clusters (**Figure 5—figure supplement 2A**) while the knockdowns exhibit various levels of disorganization (**Figure 5—figure supplement 2B–H**). Since the endogenous small micromeres normally home over a very short distance, the knockdown small micromeres, as might be expected, remain close to the coelomic pouches—they do not disperse from that vicinity, but they do not tightly cluster in the correct location. Also, at this time, there is

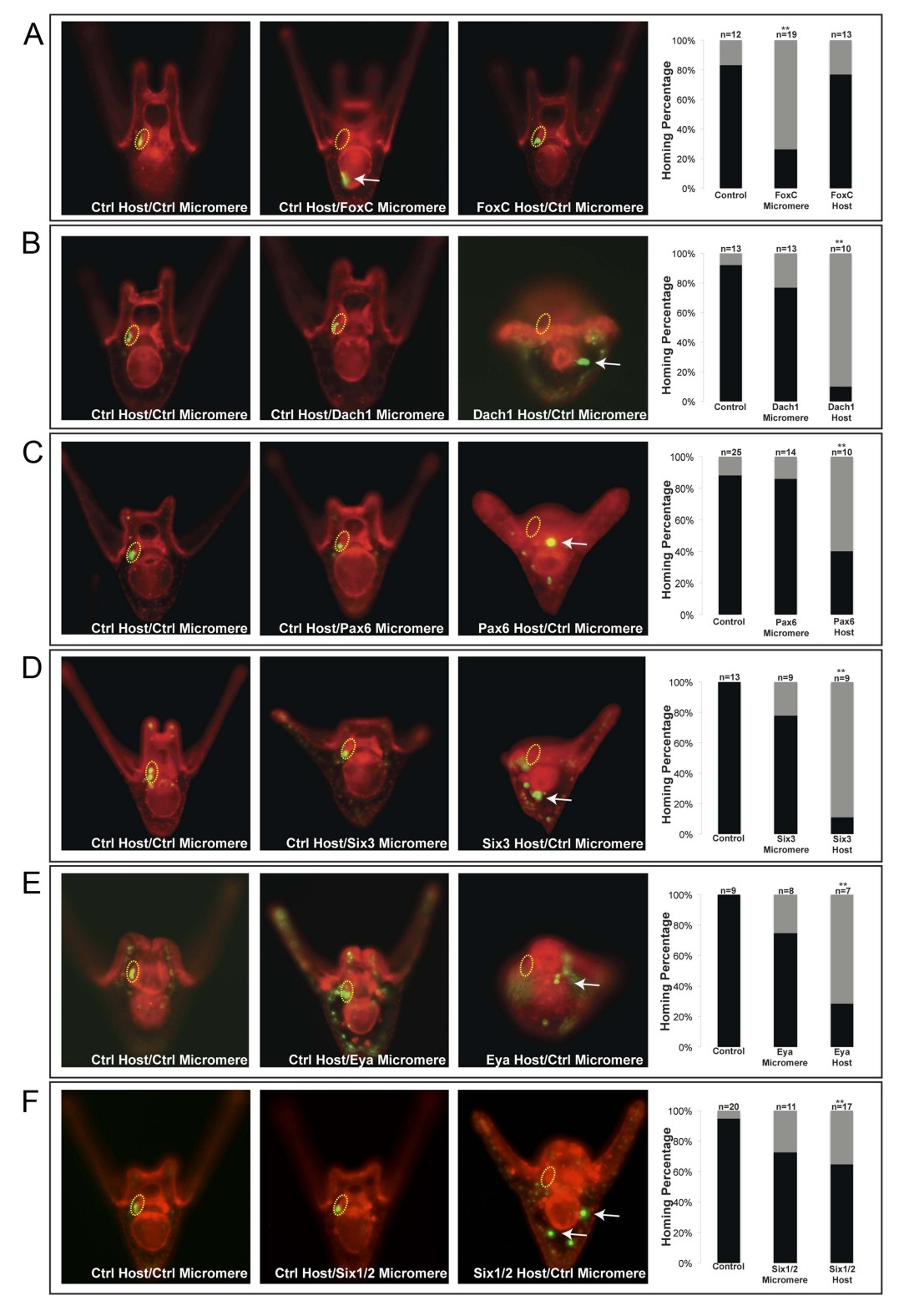

**Figure 5**. Homing is controlled by upstream transcription factors. By selectively knocking down specific transcription factors in either an ectopically placed micromere or the host embryo, transcription factors for the homing were identified. (**A**) FoxC, a transcription factor expressed in the small micromeres and oral mesoderm was necessary for homing of the small micromere (n = 19, p < 0.001), but had no inhibitory effect on homing when knocked down in just the mesoderm (n = 13). (**B**) Dach1, an aboral mesoderm transcription factor, affected homing when knocked down in the mesoderm (n = 10, p < 0.0001) but did not affect homing when perturbed in the small micromere (n = 13). (**C**) Pax6, affected homing when knocked down in the mesoderm (n = 10,

*Figure 5. continued on next page*

*Figure 5. Continued*

p < 0.004) but had no effect on homing when perturbed in the small micromere (n = 14). (**D**) Aboral mesoderm transcription factor, Six3 affected homing when knocked down in the mesoderm (n = 9, p < 0.00009) but had no effect homing when perturbed only in the small micromere (n = 9). (**E**) Aboral transcriptional co-activator, Eya, affected homing when knocked down in the mesoderm (n = 7, p < 0.002) but did not affect homing when perturbed in the small micromeres (n = 8). (**F**) Aboral transcription factor, Six1/2, affected homing when knocked down in the mesoderm, as well (n = 17, p < 0.02) and did not affect homing when knocked down in the micromere (n = 11). Yellow circles indicate the location of the coelomic pouch. White arrows indicate 'lost' micromeres. The graph depicts the percentage of embryos (Control Host vs MO micromere vs MO Host) seen with the given ability to home (Black-Homing vs Gray-No Homing), 'n' equals the number of embryos scored in each case p-values were calculated using a $\chi^2$ test. '**' denotes a statistically significant (p < 0.05) p-value.

The following figure supplements are available for figure 5:

**Figure supplement 1**. Coelomic pouch transcription factors that do not affect homing.

**Figure supplement 2**. Whole embryo perturbation of coelomic pouch transcription factors assayed for *vasa* expression.

prevalent cell rearrangement at the tip of the archenteron that leads to the fusion of the oral ectoderm and foregut endoderm (unpublished observations).

Thus of the eleven genes screened for homing behavior, only six were necessary for homing. We next wanted to better understand how the five coelomic pouch mesodermal transcription factors scored as necessary for homing might relate in the network circuit that runs in that tissue.

## Perturbation analysis unveils a GRN circuit that controls homing

Each of the five transcription factors involved in controlling homing is expressed in the aboral coelomic pouch. Luo and Su showed that transcription factors Dach1, Eya, Six1/2, and Pax6 are co-localized in the aboral coelomic pouch (*Luo et al., 2012*). Six3 was found to be expressed in the aboral mesoderm early in development, and by double fluorescent in situ hybridization, we see *six3* to also overlap with aboral marker, *eya* (*Figure 6B*) (*Wei et al., 2011*). *Six3* was seen to be tightly apposed with *vasa* expression but is exclusive from small micromeres. Also, *six3* expression is distinct from the oral, *myosin,* domain of expression (*Figure 6A,C*). Expression domains within the coelomic pouch are illustrated in *Figure 6D*. Luo and Su also noted that transcription factors Dach1 and FoxC are expressed in the oral coelomic pouch (*Luo et al., 2012*). Transcription factor FoxC, however, is expressed in both the oral coelomic pouch and the small micromeres at the time of small micromere homing (*Luo et al., 2012*).

To determine the order of expression of the transcription factors that could potentially act in a homing subcircuit, we first performed whole mount in situ hybridization on all five homing genes (*dach1*, *pax6*, *eya*, *six1/2*, and *six3*) at eight time points ranging from pre-hatched blastula (before small micromere migration along the archenteron) to early pluteus (after the small micromeres and coelomic pouches have coalesced) with a 2- to 4-hr step-time to establish relative timing of expression (*Figure 6—figure supplement 1*). By surveying this gene set, we conclude that their co-localization in the coelomic pouches and temporal expression patterns suggest that these transcription factors may be interacting (diagram, *Figure 6D*). To ask how or if these genes interact in a gene network to drive the morphogenetic cassette, we undertook a perturbation analysis to build a homing GRN upstream of the morphogenetic movement.

Each transcription factor was perturbed and the effect of that perturbation on the others was examined. When Dach1 was perturbed using a morpholino specific to the start of translation, we saw by in situ hybridization an increased expression of *dach1* (indicating self repression). Knowing that *dach1* is expressed in the endoderm earlier in development from our time course data, we concluded that the upregulation seen in the gut was a result of *dach1* self-repression not only in the gut but also later in the coelomic pouch mesoderm expression. A down-regulation of *six3* and a down-regulation of *eya* transcripts was also seen in the Dach1 perturbation. These data indicate that Dach1 positively regulates *six3* and *eya* and inhibits the overexpression of itself by repression (*Figure 7*).

When Six3 was perturbed using a start site morpholino, we saw a self up-regulation of *six3* but only in the apical expression domain, a down-regulation of *eya*, and a down-regulation of *pax6* transcripts.

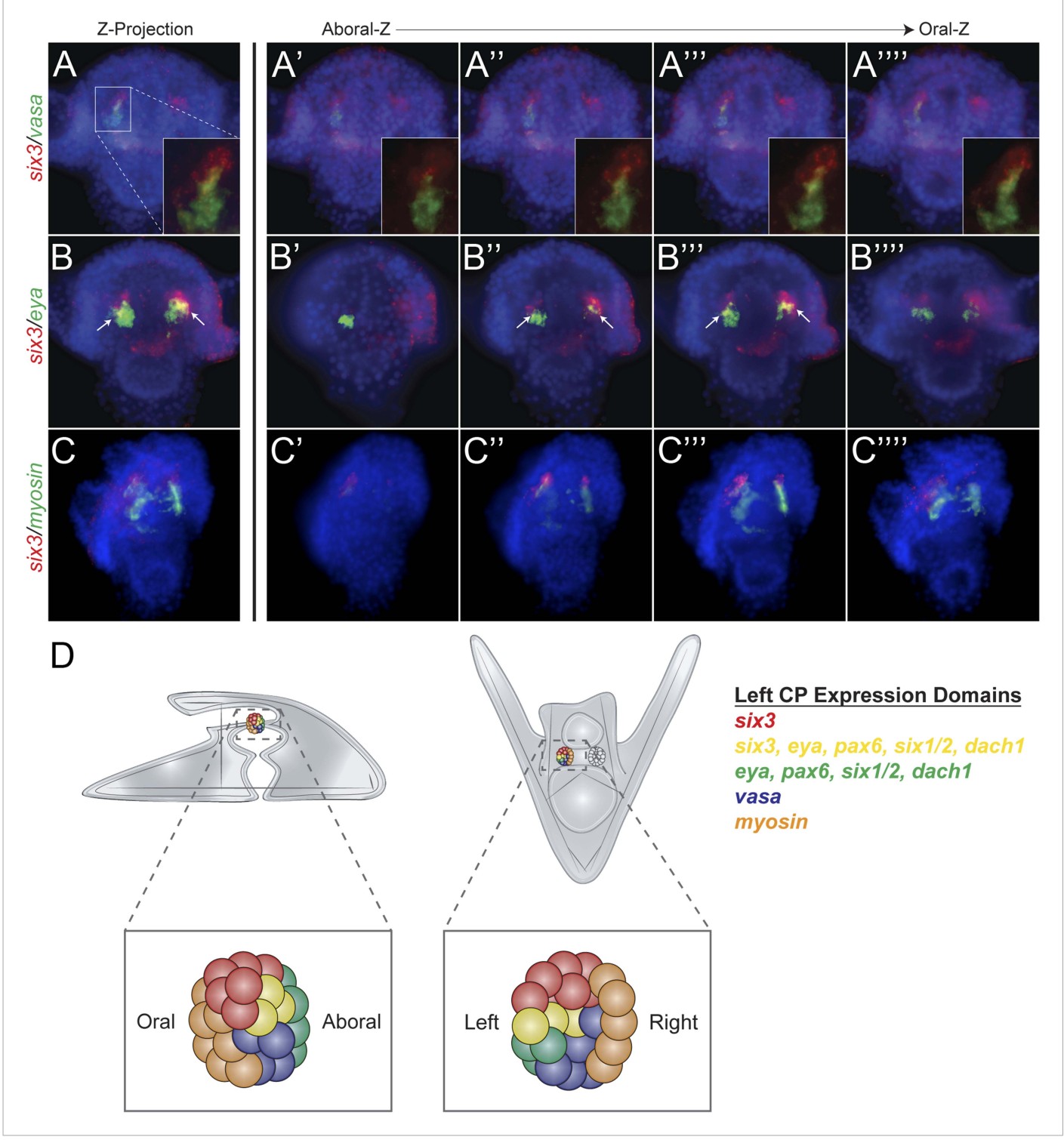

**Figure 6**. Expression domains within the coelomic pouch. Double fluorescent in situ hybridization of *six3* with *vasa* shows a tight apposition of their expression domains but a lack of co-localization (**A–A''''**). Z projection of *six3* and *vasa* is seen in (**A**), and individual Z sections are seen from aboral to oral most locations in the embryo in (**A'–A''''**). Insets on panels **A–A''''** show a zoomed perspective of just the left coelomic pouch apposed expression of *six3* and *vasa*. *six3* and *eya* were seen to overlap in the aboral coelomic pouch in both the Z projection (**B**) and Z sections (**B''** and **B'''**) but not in Z sections (**B'** or **B''''**). *myosin* expression was seen to be in the most oral expression domain of the coelomic pouch, distinct of *six3* (**C–C''''**). An illustration demonstrating the expression domains observed in the coelomic pouch from our data and the data of Luo and Su, 2012 is displayed in (**D**) from both the lateral and oral views.

*Figure 6. continued on next page*

*Figure 6. Continued*

The following figure supplement is available for figure 6:

**Figure supplement 1**. Spatial and temporal expression of homing genes by whole mount in situ hybridization.

We conclude from these data that Six3 positively regulates *eya* and *pax6* in the coelomic pouch mesoderm. Perturbing Eya in the same way showed a down-regulation of *six3*, *six1/2*, and a self-down-regulation of *eya*. Knowing that Eya is a transcriptional co-activator of Six1/2, Eya must positively regulate itself, *six3*, and *six1/2* by co-acting with Six1/2. Next, Six1/2 was perturbed, and, as expected, regulatory interactions found reflected those found in the Eya knockdowns indicating that Eya and Six1/2 do, in fact, act synergistically as has been seen before (*Heanue et al., 1999*; *Materna et al., 2013*). Finally, Pax6 perturbation caused down-regulation of *six3*, *eya*, and itself. Pax6, therefore, positively regulates *six3, eya*, and itself (*Figure 7*).

When the network was laid out in a Biotapestry model, the subcircuit controlling the homing mechanism was readily discerned as a coherent feed-forward loop composed of multiple feedback loops that sustain its robust nature (*Figure 8A*) (*Longabaugh et al., 2005*). It also displayed a striking resemblance to another circuit well known in development: the RDGN. The Pax-Six-Eya-Dach RDGN is necessary for specification of retinal cell types in many systems (*Purcell et al., 2005*; *Kozmik et al., 2007*; *Kumar, 2009*; *Salzer et al., 2010*; *Weasner and Kumar, 2012*). The same five factors contribute to a coherent feed-forward circuit for *Drosophila* eye specification (*Kumar, 2009*) (*Figure 8B*). Few linkages between the eye and small micromere homing circuits are different (*Figure 8*).

## Discussion

Our chimera experiments show that small micromeres home with high fidelity to the coelomic pouches no matter where they initially are placed in the embryo. That homing utilizes an autonomous EMT and a directed movement to the target site in the coelomic pouch. Results detail some of the complex morphogenetic movements of the small micromeres in their migratory trajectory. Perturbation analysis identified the transcriptional inputs upstream of the putative homing signal. Network analysis of the transcription factors necessary for homing signal production unveiled a stereotypical coherent feed-forward subcircuit that controls the attractive signal. That circuitry shows a striking similarity to the RDGN or Pax-Six-Eya-Dach Network.

That the same subcircuit is present and used for another purpose in *Drosophila* was an unexpected outcome. While it is possible that the similarity is due to convergent evolution, deployment of the same subcircuit in the two organisms for two very different purposes also suggests another possibility. Ancient subcircuit functions necessary for systems-level operations in networks may be employed as a unit as new networks evolve rather than build new systems level operations gene by gene. It isn't yet known whether evolution favors subcircuits of similar topology moving as a unit to perform similar functions in diverse cell types, but advances in systems-level studies should reveal the frequency of this phenomenon.

### Small micromere homing is a complex directed cell migration mechanism

The data from the chimera experiments show that normally small micromeres retain an adhesion to adjacent cells during archenteron extension. When they arrive at the tip of the archenteron, they then leave the epithelium using an EMT through a laminin-containing basement membrane, migrate to the posterior end of the coelomic pouch, and transition back to an epithelium as they coalesce with the coelomic pouch. Ectopically placed small micromeres behave differently from the endogenous small micromeres in that they fail to retain epithelial adhesions through gastrulation and instead go through an EMT at the time PMCs ingress. The fact that the ectopically placed small micromeres lose adhesion and go through EMT independent of the timing of archenteron tip formation implies that there must be a non-autonomous cue to keep the endogenous small micromeres in place during gastrulation. Without that local cue, the small micromeres actively home from the beginning of gastrulation as seen with the behavior of ectopically placed small micromeres. Further, there must be a non-autonomous signal that is recognized by small micromeres when they reach the tip of the forming archenteron that cues them to initiate the

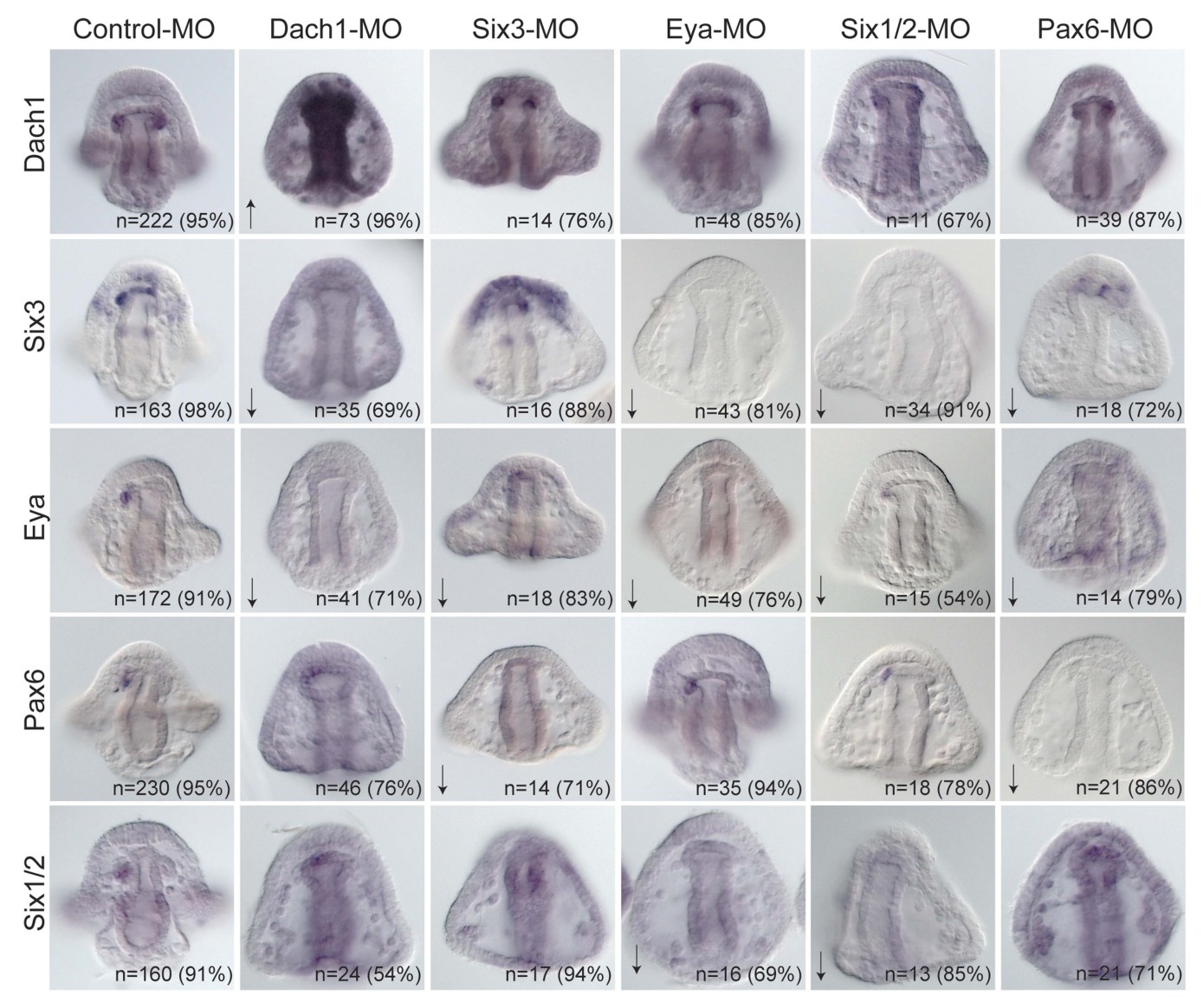

**Figure 7**. Perturbation analysis unveils regulatory linkages in a homing GRN. Perturbations using a Dach1 morpholino (MO) showed Dach1 to be upstream of *dach1*, *six3*, and *eya*. A Six3-MO caused a downregulation of *eya* and *pax6*. Pertubations with an Eya-MO showed Eya to be upstream of *six3*, *six1/2*, and *eya*. Six1/2 is seen to be upstream of *six3*, *eya*, and *six1/2*. Finally, a Pax6-MO caused a downregulation of *six3*, *eya*, and *pax6*. Arrows represent up or downregulation seen for each panel. 'n' equals the total number of embryos scored, and the adjacent percentage designates the percent of embryos scored with the shown effect. Morpholino morphology phenotypes at 24 hpf and 48 hpf are presented in *Figure 7—figure supplement 1*. Unpublished morpholino data has been validated using a second, distinct morpholino (see *Figure 7—figure supplement 2*).

The following figure supplements are available for figure 7:

**Figure supplement 1**. Morpholino phenotypes.

**Figure supplement 2**. Morpholino effects seen in this paper that have not previously been validated were confirmed using a second morpholino designed in a location distinct to the site of the first morpholino.

EMT and migration. The behavior of the ectopic cells suggest that the migratory capacity can be activated at the beginning of gastrulation, but since they normally do not move from the epithelium until at the tip of the archenteron, another event may stimulate them to complete their journey to the coelomic pouches.

**Figure 8**. Homing GRN subcircuit shows striking resemblance to *Drosophila* RDGN. (**A**) The perturbation analysis was mapped as a Biotapestry network model. (**B**) A retinal gene network subcircuit extracted from Drosophila (*Kumar, 2009*) shows a very similar circuit with few regulatory linkage changes in comparison.

The following figure supplement is available for figure 8:

**Figure supplement 1**. The RDGN subcircuit is conserved throughout evolution.

Observations of ectopically placed small micromeres also reveal that the cells undergo directed cell migration to get to the coelomic pouch mesoderm. The cells find their way to the coelomic pouch no matter their starting location. In other systems cells become polarized in response to a chemoattractant signal at the leading edge of the cell while the remainder of the cell is cued to receive inhibitory or repulsion signals responsible for increasing its sensitivity to its target location (as reviewed by *Richardson and Lehmann, 2010*; *Reig et al., 2014*). Small micromeres appear to take a fairly direct route to their eventual target no matter where they are initially placed. Since the normal route involves movement through part of the blastocoel, and since ectopic small micromeres also use the blastocoelar route, the putative homing signal likely is produced earlier than the end of gastrulation since the ectopic small micromeres move through the blastocoel to the correct target well before gastrulation is completed.

At present, it is only speculation that there is a chemoattraction mechanism at play. If it were chemoattraction, the cue must work at the individual cell level since single small micromeres respond to the cue when ectopically placed. The transcriptional regulation upstream of that putative signal offers approaches for its identification. Potentially, as an alternative hypothesis, the coelomic pouch NSM could be drawing the small micromeres to their final site by filopodia, since it is known that these cells as well as the small micromeres extend filopodia (*Hardin, 1988*; *Campanale et al., 2014*). These hypotheses will need to be explored further.

## Control of morphogenetic movements at a systems level involves a series of subcircuits to drive complex cell behavior

The goal of determining upstream transcriptional regulation of the homing process requires connection of the aforementioned network (*Figure 8A*) to chemoattraction mechanisms (both attractant and repulsion) directly responsible for the directing migration. In addition, other subcircuits are necessary to drive the EMT, for reception of the putative signal, and for directed movement in response. Transcriptional control of directed cell migration is not well studied at a systems level. There are a few examples, however, that suggest the way these systems work. Findings in the tunicate, *Ciona intestinalis*, describe the distinct transcriptional regulation of a cellular module that drives migration of heart precursors. The heart GRN drives membrane protrusions and RhoGTPase signaling, tail retraction, adhesion, and polarity (*Christiaen et al., 2008*). One important feature of that study was that the cells constitutively expressed many of the factors required for these cell behaviors. It only took one or two factors to be under direct control of the developmental GRN to regulate the timing, directionality, onset, and termination of the heart cell precursors' journey. In the sea urchin, five different GRN subcircuits were shown to drive five different cellular components of an EMT (*Saunders and McClay, 2014*). It is highly likely, like *Ciona* heart cells, that the sea urchin developmental GRN will control directly expression of only a fraction of the proteins used for de-adhesion, motility, invasion of the basement membrane, altered cell

polarity, and cell shape changes, all components of an EMT, while many other proteins in those processes will be constitutively expressed. Using small micromere homing as an assay, we were able to begin to unravel the intricacy of GRN subcircuits that control cell migration to a target site. This work thus demonstrates how morphogenesis can be controlled by conserved specification GRN subcircuits.

The small micromere homing example reveals something else in morphogenetic movement control. We tend to think of the result of the movement: the small micromeres become associated with the aboral coelomic pouch through homing. Yet, to accomplish that feat, both the cells that home and the target cells must enact a series of coordinated, but independent processes. Here, we found a set of genes that control the production of the putative homing signal. Being that the subcircuit is of coherent feed-forward topology, we suspect it will be found at the most distal part of a larger GRN; therefore, it will act downstream of the specification circuitry of the signal-producing cell type. Connecting that upstream subcircuit to the downstream signal(s) will provide a tool for explaining one component of the homing mechanism that we know to be transcriptionally controlled.

## Deployment of a whole subcircuit controls distinct developmental processes

Recognizable RDGN components are deployed in a variety of developmental processes in different organisms. After pooling published data from three different fields of study (demosponge canal systems, vertebrate skeletal muscle, and mouse otic placodes), we were able to construct putative GRN models (*Figure 8—figure supplement 1*). Although each respective subcircuit is embedded in a larger specification GRN, when extracted as a subcircuit, it appears that the RDGN feed-forward topology has been deeply conserved (*Torres et al., 1996*; *Heanue et al., 1999*; *Relaix and Buckingham, 1999*; *Xu et al., 1999*; *Ridgeway and Skerjanc, 2001*; *Kardon et al., 2002*; *Laclef et al., 2003*; *Riley and Phillips, 2003*; *Zheng, 2003*; *Ozaki, 2004*; *Silver, 2004*; *Relaix et al., 2013*; *Rivera et al., 2013*). Only now that we have been able to make a complete RDGN subcircuit in the sea urchin, can we hypothesize about what evolutionary mechanism may be at play. With the same transcription factors involved in a similar feed-forward process in such a variety of tissues, we hypothesize that the RDGN subcircuit is ancient and has been repeatedly deployed as a unit to provide feed-forward control.

If this subcircuit is redeployed throughout evolution as a unit, as seems to be the case given the frequency of its presence, the mechanism that keeps it intact is unknown. Evolving as a modular unit must mean that the subcircuit serves as an indispensible or important component in a network full of modular components. The assembly of the coherent feed-forward circuit, once established, appears to remain intact in separate lineages for the same reason: that circuit must be critical for the GRNs in which it is embedded. Its presence raises the question of whether there are other ancient or more recent subcircuits that have evolved as modules. Since analysis of developmental GRNs is a relatively young field, it is possible that a number of ancient modular subcircuits evolve as units to assist in the diversification of cell types and morphogenesis. One obvious outcome of this mode of inheritance would be a contribution to robustness in developmental mechanisms. And perhaps once a subcircuit is particularly well integrated and suited for a given type of regulatory function, each of its members (in our case, transcription factors) might act as a constraint on the others because any reduction in integration would be less adaptive.

What is intriguing is how ancient the RDGN must be. It is a circuit that is used because it efficiently provides an excellent feed-forward circuit that stabilizes the process to which it is attached. It is best studied in retinal determination but its ancestral role is unknown and could equally well be directed cell migration.

## Materials and methods

### Adult animals and embryo culture

Adult *Lytechinus variegatus* were obtained from Reeftopia (Key West, FL, United States) or the Duke University Marine Lab (Beaufort, NC, United States). Gametes were obtained by injection of 0.5 M KCl into the adult coelom. Embryos were cultured at 23°C in artificial seawater (ASW).

**Table 1**. Transplant efficacy for microsurgeries

| Microsurgery | Successful transplants (%) | Failed transplants (%) |
|---|---|---|
| Ctrl host/Ctrl micromeres (*Figure 5*) | 51 ± 9 | 49 ± 9 |
| Ctrl host/MO micromeres (*Figure 5*) | 40 ± 20 | 60 ± 20 |
| MO host/Ctrl micromeres (*Figure 5*) | 48 ± 17 | 52 ± 17 |

## Cloning of RDGN genes

The full-length coding sequences for NSM transcription factors, Delta, and RDGN genes were obtained by designing primers against a transcriptome data set. Nucleotide sequences were annotated and deposited on GenBank (Accession Numbers: *Table 2*).

## Morpholino and mRNA microinjections

Morpholino antisense oligonucleotides were designed against the start site of translation by GeneTools. Morpholino sequences and concentrations are as listed in *Table 3*. Morpholinos were diluted in molecular-grade $H_2O$ and either FITC or TMR injectable dyes. Morpholino experiments were carried out on three biological triplicates, cultured at 23°C, and assayed by in situ hybridization. Phenotypes of morpholinos at 24 hpf and 48 hpf were imaged to show the overall health and morphology of knockdowns (*Figure 7—figure supplement 1*). Published morpholinos listed in *Table 3* were verified using appropriate controls explained in the citation referenced. Two morpholinos were assayed to test the specificity of Six3, FoxF, and FoxC. Appropriate efficacy controls were also carried out previously for Six3, FoxF, FoxC, Delta, Six1/2, and Dach1 as is cited in *Table 3*. Previously unpublished morpholinos were validated by ordering a second, distinct morpholino to the start of translation or 5′ UTR and were verified using in situ hybridization (*Figure 7—figure supplement 2*). mRNA for injection was transcribed in vivo using Ambion mMessage mMachine. Concentrations for mRNA injections: 300 ng/µl Histone2B-GFP, 500 ng/µl Histone2B-RFP, 500 ng/µl membrane-RFP, and 300 ng/µl membrane-GFP. For injection, mRNAs were diluted in 20% glycerol in diethylpyrocarbonate-treated $H_2O$.

## Microsurgeries

At 2.5 hpf, one micromere from an injected donor embryo was transplanted onto a differentially injected 16-cell host embryo in the place of one discarded host micromere or at an ectopic location. Microsurgery was performed with fine glass needles and Narishige micromanipulators. Detailed methods of injections and transplants were followed as previously described (*Logan et al., 1999*). Transplant efficacy was scored in *Table 1*.

**Table 2**. Plasmids used in this study

| Plasmid | Fragment | Accession number |
|---|---|---|
| Dach1 | Full CDS | KR181947 |
| Delta | Full CDS | KR181946 |
| Eya | Full CDS | KR181945 |
| FoxC | Full CDS | KR181944 |
| FoxF | Full CDS | KR181943 |
| Pax6 | Full CDS | KR181942 |
| PitX2 | Full CDS | KR181941 |
| Six3 | Full CDS | KR181940 |
| SoxE | Full CDS | KR181939 |

## Live image acquisition

At 11 hpf, embryos were de-ciliated in 2× hypertonic artificial seawater, mounted in 1× artificial seawater containing 10 µM p-methoxy-phenyl isoxazoline (*Semenova et al., 2011*) on a slide coated in 1% protamine sulfate and sealed with a combination of Vasoline, lanolin, and paraffinwax (also known as V.A.L.A.P.). Images were acquired using Coolsnap high-resolution CCD camera on the DeltaVision Elite with 40×/0.65–1.35 Oil UAPO40X0I3/340 DIC objective. Images were collected at 3-min intervals beginning at 12 hpf and ending at 18 hpf or later and projected as videos using SoftWorx for DeltaVision.

**Table 3**. Morpholino anti-sense oligonucleotide sequences used in this study

| MASO | MASO sequence | Morpholino type | Working concentration |
|---|---|---|---|
| LvPax6-MO1 | GTTGACCTGGCATAGCAGCATTTAC | Translation blocking | 1.2 mM |
| LvPax6-MO2 | CATGTCCCCGTGACCCATAGTTTTC | Translation blocking | 0.3 mM |
| LvEya-MO1 | GCGCTGAAGCTATTTGACATGCTGT | Translation blocking | 0.75 mM |
| LvEya-MO2 | GTTGAAACCTGTTTGACTGTAGGCC | Translation blocking | 1 mM |
| LvSix3-MO | ATGTTTCCGACTCCGTCCAAACCAT | Translation blocking (*Wei et al., 2009*) | 0.75 mM |
| LvSix1/2-MO | CCCAAGTCCGTGGCAAGGATAAGAT | Translation blocking (*Ransick and Davidson, 2012*) | 0.5 mM |
| LvDach1-MO | AGTAGGCGGTGGACTTCCCATTTTC | Translation blocking (*Peter and Davidson, 2011*) | 0.5 mM |
| LvFoxC-MO | TGAAGCGTACATTGGCATGGATGTT | Translation blocking (*Andrikou et al., 2015*) | 0.75 mM |
| LvDelta-MO | GTGCAGCCGATTCGTTATTCCTTT | Translation blocking (*Sweet et al., 2002*) | 0.4 mM |
| LvFoxF-MO | TCTAATTGAGTCATCTGGAGAGTGT | Translation blocking (*Andrikou et al., 2015*) | 0.75 mM |
| LvSoxE-MO | GCTCTAAACTCTCAGGGCTACTCAT | Translation blocking | 0.75 mM |
| LvPitX2-MO | ACTGGTTCATCGCTGCTGATTAATT | Translation blocking | 0.6 mM |
| Control-MO | CCTCTTACCTCAGTTACAATTTATA | Translation blocking | ExptDependent |

## Immunostaining and fixed imaging

Embryos were fixed in 4% paraformaldehyde for 10 min, washed in PBST, blocked in 4% normal goat serum (in PBST) for 45 min, and incubated in polyclonal anti-Laminin antibody (1:250) (ab11575; Abcam [Cambridge, MA, United States]) and anti-GFP (1:2000) (ab13970; Abcam) overnight at 4°C. Embryos were then washed in PBST, incubated in a 1:200 dilution of either Cy2- or Cy3-conjugated secondary antibody, and finally, stained with Hoechst's (1:1000) (Molecular Probes [Eugene, OR, United States]) to label all nuclei. Images were acquired using Zeiss LSM 510 upright confocal with a 40×/1.4 Oil Plan-Apochromat objective. Z-slices were spaced 1.0 μm apart spanning the diameter of the embryo.

## Whole mount in situ hybridization

RNA in situ hybridization was performed using Digoxigenin-11-UTP-labeled probes as previously described (*McIntyre et al., 2013*). Briefly, embryos were fixed in 4% paraformaldehyde overnight at 4°C, washed with ASW, and stored in methanol at −20°C. RNA probes were synthesized in vitro and used at 1 ng/μl. Hybridization took place at 60–65°C. Probes were visualized using alkaline phosphatase-conjugated anti-DIG antibody (1:1500, Roche [Indianapolis, IN, United States]). Finally, color was developed using NBT/BCIP (Roche). For double fluorescent in situ hybridization, a second probe (labeled with Fluorescein-12-UTP) was hybridized. Expression was visualized using the Tyramide Signal Amplification system (TSA-plus kit, Perkin Elmer [Waltham, MA, United States]). Embryos were visualized with a Zeiss Axioplan2 upright microscope.

## Statistical analysis

GraphPad software was used to analyze a 2 × 2 contingency table with rows corresponding to the condition (controls and knockdown micromeres or controls and knockdown hosts) and columns corresponding to outcomes ('homing' or 'no homing'). Chi-squared was then calculated, and a p-value was declared significant at a value of less than 0.05.

## Acknowledgements

We would like to thank members of the McClay lab and Dr Ryan Range for providing insightful feedback on the manuscript. Authors would also like to acknowledge the Gene Regulatory Networks for Development course at MBL for the stimulating discussions about evolution of gene networks, in particular, the RDGN. Support for this project was provided by NIH RO1-HD-14483 and NIH PO1-HD-037105.

## Additional information

### Funding

| Funder | Grant reference | Author |
| --- | --- | --- |
| National Institute of Child Health and Human Development (NICHD) | NIH RO1-HD-14483 | David R McClay |
| National Institute of Child Health and Human Development (NICHD) | NIH PO1-HD-037105 | David R McClay |

The funder had no role in study design, data collection and interpretation, or the decision to submit the work for publication.

### Author contributions

MLM, DRM, Conception and design, Acquisition of data, Analysis and interpretation of data, Drafting or revising the article

---

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
