## [Decision Letter]

Thank you for submitting your work entitled “Deployment of a Retinal Determination Gene Network Drives Directed Cell Migration in the Sea Urchin Embryo” for peer review at *eLife*. Your submission has been favorably evaluated by Aviv Regev (Senior Editor), Marianne Bronner (Reviewing Editor), and three reviewers.

The reviewers have discussed the reviews with one another and the Reviewing editor has drafted this decision to help you prepare a revised submission

This is an interesting and valuable paper that examines the “homing” of small micromeres, which are likely primordial germ cells, to the coelomic pouches in sea urchins. The authors use elegant microsurgical approaches to demonstrate this homing behavior and to show its dependence on various transcription factors expressed in the coelomic pouches. Based on these genes and the regulatory interactions among them, the authors show that a genetic circuit previously associated mostly with retinal determination in *Drosophila* also functions in the coelomic pouches to direct the migration of sea urchin small micromeres. The paper makes an important contribution and should be published after certain revisions:

Essential revisions:

1) Circuit diagrams: There's a logical problem in the circuit diagrams shown. The diagrams indicate auto-repression of Dach1 and Six3. But the WMISH data show that the effect of Dach1 and Six3 knockdown is to activate expression of these genes in tissues other than coelomic pouches (Dach1 expression is activated in the gut and Six3 is up-regulated in the animal pole ectoderm). If Dach1 and Six3 are normally expressed only in the coelomic pouches, this must be a non-cell-autonomous effect. In any case, it's occurring in tissues other than coelomic pouches, and the circuit diagrams are misleading in that they represent an amalgamation of regulatory interactions that are not really occurring in the same cells. Also, since the coelomic pouches are a mixture of cell types, has it been shown clearly that Dach1, Six3, Eya, and Pax6 are expressed in the same cells of the pouch?

2) Quantification of effects on gene expression: Since some of the effects of the morpholino knockdown appear to be partial, based on the qualitative WMISH images shown (e.g., the effect Pax6 KD effect on Six3 expression), it would be valuable to confirm these qualitative results with QPCR. In some cases, no regulatory linkage is shown where it seems there may be an effect, based on the qualitative WMISH data (e.g., Pax6 expression seems somewhat fainter in the *eya* morphants than in controls). I realize that QPCR is problematical for genes that are expressed in multiple tissues if the effect one is interested in is restricted to just one tissue, but some of the genes studied seem to be expressed primarily in the coelomic pouches (at least at some stages) and could therefore be analyzed by QPCR or other quantitative methods.

3) Controls for morpholinos: What controls were carried out to confirm the specificity of the MOs? Two different MOs are listed for Pax6 and Eya, but only one for all others. For Pax6 and Eya, which specific experiments/data were based on MO1 and which were based on MO2? The authors should also include some DIC images showing the general morphology of their morphants, to help readers judge the health of the embryos and the possibility of non-specific effects. In Figure 5, the *eya*, *six3*, and *dach1* morphants look as if they did not extend arms normally. Also, since these are translation-blocking MOs and there are no antibodies against the target proteins that can be used to assess MO effectiveness, the authors need to avoid statements based on negative results (lack of an effect), e.g., in the subsection “Coelomic pouch transcription factors affect small micromeres’ ability to home”: “Thus of the ten genes screened for homing behavior five were necessary and five had no effect if missing.”

4) Details of the anatomy: Where exactly within the coelomic pouches do the small micromeres end up? The posterior region? The aboral region? A diagram would have been very helpful here. What is their final position relative to the domains of expression of Dach1, Six3, Eya, and Pax6? Where are they positioned relative to the presumptive muscle cells? A more detailed description of the anatomy would have been very helpful, if these details are known.

5) The section of the Discussion that deals with circuit conservation/evolution is loose and speculative, and could be shortened.

6) The homing behavior of small micromeres is intriguing and may provide a great model to study cell migration. However, one major curiosity is whether the developmental mechanism used by the ectopic small micromeres is also used by endogenous small micromeres when they migrate during normal embryogenesis. For example, it is important to check whether any migration defects can be observed in endogenous small micromeres after Delta, Foxc, Dach1, Pax6, Six3, or Eya is knockdown (as in Figure 4 and Figure 5).

7) There is concern is the expression pattern of *six3*. In the legend of Figure 6—figure supplement 1, the authors described that: “*six3*'s earliest detectable expression is at hatched blastula […] in the apical plate domain and the future coelomic pouch cells.” However, in Figure 6—figure supplement 1, *six3* expression is almost undetectable in blastula and gastrula stage (B1-B4) and this pattern is very different from a published work by Wei et al. (Development 136, p1179, 2009). Wei et al. clearly showed that the *six3* gene identified from the California purple sea urchin is first expressed in the animal pole domain at the early blastula stage. The expression in the animal pole can be observed throughout gastrulation in addition to a second expression domain “in some secondary mesenchyme cells scattered throughout the blastocoel and at the tip of the archenteron.” In the subsection “Perturbation analysis unveils a GRN circuit that controls homing”, the authors mentioned “*six3* was found to be expressed in the aboral mesoderm”, and cited [43]. However, whether *six3* is expressed in the oral or aboral coelomic pouches or even in the small micromeres requires double staining of *six3* with appropriate markers. Where *six3* is expressed is important for interpretation of the data. In Figure 5, it is intriguing that knockdown of *six3* in both micromeres and the host affected homing. Therefore, it is possible that *six3* is expressed in both small micromeres and the coelomic pouch mesodermal cells. One additional issue is that Luo and Su (PLoS Biology, 2012) showed that in addition to the three factors studied by the authors (*dach1*, *eya* and *pax6*), *six1/2* is also expressed in the aboral coelomic pouch. I wonder whether the authors have investigated this obvious and putative player of the conserved RDGN network.

8) It is not clear the labeled red cells are migrating to their final site, or are drawn by static cells. This author showed some time ago that the embryo has very long processes that extend across the blastocoel even. Better documentation of the movement of the red cells is needed. What is present appears like a straight line – as might be suggested by a retracting process from a static cell instead of migration. The authors claim that the small blastomeres are not germ cells, or primordial germ cells, because the lineage of the line has not been documented. They conclude that the cells are multipotent but fail to provide such evidence needed for this conclusion.

9) Do any cells die? Better quantitation is necessary and they need to document cell death.

10) How can the authors distinguish if the MO is specifically testing homing, versus aspects of earlier development?

11) The methods for statistical analysis need to be described.

12) Issues with figures:

Figure 3—figure supplement 1 is confusing. The ectopic cell is labeled in red in 1A. However, it is not clear whether the red cells migrated into the coelomic pouches. An indication of coelomic pouches or counter stained the embryos with DAPI would help to see the location of coelomic pouches. Similar problem in B. The ectopic cell is now in green. I see there are two cells with green membrane and red nuclei. Are they ectopic or endogenous cells?

Figure 2, it is not clear whether a micromere or a small micromere was transplanted? If a micromere is transplanted, how can the authors be sure the two cells breached the laminin layer (indicated with a yellow arrow) at 10 hpf are small micromeres but not large micromeres?

The authors provided data to validate unpublished morpholinos using a second morpholino against Eya and Pax6 in Figure 6–figure supplement 2. Please provide references if Six3 and Dach1 morpholinos used in this study have been published and validated before.

Clear development delay is seen in the morpholino-injected embryos presented in Figure 6. This may cause problem especially in the embryos stained with the *six3* probe. Looking at the *six3* expression pattern in Figure 6—figure supplement 1B4, *six3* transcript is barely detected. At the similar stage (based on their similar morphology) showed in Figure 6, no expression of *six3* is interpreted as downregulation. Please clarify the morpholino effect on developmental delay and their effects on *six3* expression at a comparable developmental stage. Other issues concerning Figure 6: (1) *six3* expression expands in *six3* morphants, but the expansion is mostly seen in the apical plate. Therefore, the self-repression is in the apical plate not the coelomic pouches; (2) *six3* expression seems upregulated in the hindgut of the *eya* morphants. Please confirm whether it is a consistent result.

Figure 3 presented percentages of the embryos showing small micromere homing. During normal embryogenesis, small micromeres migrate to the left or right coelomic pouches in a 5:3 ratio. Is there any left-right preference of the small micromere homing when they are transplanted at various positions?

Figure legend 2D: please indicate what the white and yellow lines are. Figure legend 3: (B) should be equator and (C) animal pole.

[Editors' note: further revisions were requested prior to acceptance, as described below.]

Thank you for resubmitting your work entitled “Deployment of a retinal determination gene network drives directed cell migration in the sea urchin embryo” for further consideration at *eLife*. Your revised article has been favorably evaluated by Aviv Regev (Senior Editor), Marianne Bronner (Reviewing Editor), and 3 reviewers. The manuscript has been improved but there are some remaining relatively minor issues that need to be addressed before acceptance, as outlined below. Please address the points raised by reviewers 2 and 3 in a revised manuscript:

Reviewer #2:

The authors have addressed most of my concerns and have added substantially to the manuscript. I believe the paper should be accepted for publication. I do have one small quibble: just because a MO was published previously does not mean that appropriate controls for specificity and efficacy were carried out. If so, then the authors point is well taken. If not, then the fact that someone was able to publish an experiment with the MO is irrelevant.

Reviewer #3:

The authors have answered most of my questions. One remaining issue is the exact expression domain of *six3* in the coelomic pouches. The vasa signal in Figure 6 is extremely weak. It is difficult to judge whether there is an overlapping domain between *six3* expression domain and *vasa* localization. The authors concluded that there is no overlapping and *six3* is not expressed in the small micromeres. If this is the case, the authors need to explain the defect in homing when *six3* was knockdowned in the small micromeres.

---

## [Author Response]

Essential revisions:

1) Circuit diagrams: There's a logical problem in the circuit diagrams shown. The diagrams indicate auto-repression of Dach1 and Six3. But the WMISH data show that the effect of Dach1 and Six3 knockdown is to activate expression of these genes in tissues other than coelomic pouches (Dach1 expression is activated in the gut and Six3 is up-regulated in the animal pole ectoderm). If Dach1 and Six3 are normally expressed only in the coelomic pouches, this must be a non-cell-autonomous effect. In any case, it's occurring in tissues other than coelomic pouches, and the circuit diagrams are misleading in that they represent an amalgamation of regulatory interactions that are not really occurring in the same cells.

*Dach1* is first expressed in the gut, and overexpression of *dach1* in the gut may be a result of earlier auto-repression. However, we do see overexpression in the NSM as well, so we have decided to leave the self-repression *of dach1* in the network. Six3 experiments were repeated, and the authors thank the reviewers for pointing out that the self-repression is only in the apical organ. This network linkage will be removed.

The notion that non-cell autonomous effects might be present cannot be ruled out, but since *dach1* and *six3* are both expressed in the coelomic pouch, and perturbation analyses indicate the subcircuit relationship, the simplest explanation for the circuit is that they interact autonomously within the coelomic pouches to control the homing mechanism.

Also, since the coelomic pouches are a mixture of cell types, has it been shown clearly that Dach1, Six3, Eya, and Pax6 are expressed in the same cells of the pouch?

Yes, Luo and Su, 2012 have clearly stated that *dach1, six1/2, eya,* and *pax6* are co-expressed in aboral coelomic pouch NSM, and we review their results in the subsection “Perturbation analysis unveils a GRN circuit that controls homing”. However, they did not have data on *six3* expression. These data were added (Figure 6), and it was seen that *six3* overlaps with *eya* in the aboral coelomic pouch territory. The expression data therefore clarifies any concern over the possibility of these transcription factors being able to interact.

*2) Quantification of effects on gene expression: Since some of the effects of the morpholino knockdown appear to be partial, based on the qualitative WMISH images shown (e.g., the effect Pax6 KD effect on Six3 expression), it would be valuable to confirm these qualitative results with QPCR. In some cases, no regulatory linkage is shown where it seems there may be an effect, based on the qualitative WMISH data (e.g., Pax6 expression seems somewhat fainter in the* eya *morphants than in controls). I realize that QPCR is problematical for genes that are expressed in multiple tissues if the effect one is interested in is restricted to just one tissue, but some of the genes studied seem to be expressed primarily in the coelomic pouches (at least at some stages) and could therefore be analyzed by QPCR or other quantitative methods.*

As is stated by the reviewers, the genes in our network analysis are expressed in multiple territories; none of the genes are expressed exclusively in the coelomic pouch, although they co-localize there. qPCR is not as good as in situs if the genes in question are expressed in multiple territories. Transcription factor Six1/2, while in the aboral coelomic pouch mesoderm, also exhibits expression in the pigment cell NSM (Ransick, 2012). Both *pax6* and *eya* are expressed in the aboral coelomic pouch as well as two lateral patches of apical ectoderm. We hypothesize this ectodermal territory to be a part of a later, neural (potentially retinal) network. Six3 is in the apical plate domain as well as the coelomic pouches. It is also expressed in the foregut endoderm later in development. *Dach1* is expressed earlier in the gut endoderm, and it is not until late gastrula that expression in the coelomic pouches appears. Dach1 also appears in the early pluteus at the ends of the extending arms (unpublished observations).

That being said, we acknowledge the fact that the Pax6 in situ control was a poor representation of the normal, control expression, and a better representative of all the embryos scored replaced the picture (Figure 7).

3) Controls for morpholinos: What controls were carried out to confirm the specificity of the MOs? Two different MOs are listed for Pax6 and Eya, but only one for all others.

Pax6 and Eya MO were the only unpublished morpholinos in our network; therefore, we ordered two morpholinos to test their specificity and show they had the same effect. All other morpholinos were published in the purple sea urchin (*Strongylocentrotus purpuratus*), and our morpholinos were designed to the identical sequence site. The authors would like to thank the reviewers for addressing our oversight of not adding citations for these publications. References were added to the table of morpholino sequences (Table 3).

For Pax6 and Eya, which specific experiments/data were based on MO1 and which were based on MO2?

Both morpholinos were tested for their specificity by addressing the same knockdown analysis (Figure 7), and after determining they had the same effects, they were used interchangeably for further experimentation with similar outcomes in repeats of the same experiment.

The authors should also include some DIC images showing the general morphology of their morphants, to help readers judge the health of the embryos and the possibility of non-specific effects.

Thank you, these are included in a new figure. Morphant morphology was assayed at 24 and 48 hours and is presented in Figure 7—figure supplement 1.

*In*
Figure 5*, the* eya*,* six3*, and* dach1 *morphants look as if they did not extend arms normally.*

The reviewers are right to point out that these morphants do not extend arms properly. A morpholino unrelated to this paper for Hedgehog (Hh) is also seen to have short arms (Walton, 2009). We know from the sea urchin endomesoderm GRN that Dach1 is upstream of Hh, so it would make sense that these phenotypes are similar. We have also seen that Dach1 is expressed in the early pluteus at the end of the extending arms (unpublished). Literature from other systems suggests that Eya is a co-activator of Dach1 (in addition to Six1/2) (Heanue, 1999). If this were true in the sea urchin, it would also suggest that Eya would have a similar phenotype. From our network analysis, Dach1 is also upstream of Six3. The source of the short-arm phenotype might stem from Dach1 regulation, but it is unclear, to date, what the mechanism is.

Also, since these are translation-blocking MOs and there are no antibodies against the target proteins that can be used to assess MO effectiveness, the authors need to avoid statements based on negative results (lack of an effect), e.g., in the subsection “Coelomic pouch transcription factors affect small micromeres’ ability to home”: “Thus of the ten genes screened for homing behavior five were necessary and five had no effect if missing.”

Thank you. We re-worded this sentence to now read: “Thus of the eleven genes screened for homing behavior, only six were necessary for homing”.

4) Details of the anatomy: Where exactly within the coelomic pouches do the small micromeres end up? The posterior region? The aboral region? A diagram would have been very helpful here. What is their final position relative to the domains of expression of Dach1, Six3, Eya, and Pax6? Where are they positioned relative to the presumptive muscle cells? A more detailed description of the anatomy would have been very helpful, if these details are known.

Small micromeres end their migration in the posterior coelomic pouch. We believe from our double in situ data (Figure 6) that they are preferentially located aborally within the posterior region, but we have not carried out DFISH for *vasa* with an exclusively oral coelomic pouch mesoderm marker. The *dach1, eya, six1/2,* and *six3* expression is aboral to myosin expression and anterior to the final location of the small micromeres. We have added a diagram (Figure 6) to show the territories and have added sentences to the Results section (subsection “Perturbation analysis unveils a GRN circuit that controls homing”).

5) The section of the Discussion that deals with circuit conservation/evolution is loose and speculative, and could be shortened.

Thank you. The last section of the Discussion has been shortened, and the speculation on the evolution of the network has been tightened.

6) The homing behavior of small micromeres is intriguing and may provide a great model to study cell migration. However, one major curiosity is whether the developmental mechanism used by the ectopic small micromeres is also used by endogenous small micromeres when they migrate during normal embryogenesis. For example, it is important to check whether any migration defects can be observed in endogenous small micromeres after Delta, Foxc, Dach1, Pax6, Six3, or Eya is knockdown.

We did whole embryo knockdowns for each of the transcription factors in question and stained for *vasa.* The results are presented in Figure 5—figure supplement 2. We observe that the control endogenous small micromeres are within the coelomic pouches in tight clusters while the knockdowns exhibit various levels of disorganization. Since the endogenous small micromeres home over a very short distance, the knockdown small micromeres, as might be expected, remain close to the coelomic pouches – they do not disperse from that vicinity, but they do not tightly cluster in the correct location.

*7) There is concern is the expression pattern of* six3*. In the legend of*
Figure 6—figure supplement 1*, the authors described that: “*six3*'s earliest detectable expression is at hatched blastula […] in the apical plate domain and the future coelomic pouch cells.” However, in*
Figure 6—figure supplement 1*,* six3 *expression is almost undetectable in blastula and gastrula stage (B1-B4) and this pattern is very different from a published work by Wei et al. (Development 136, p1179, 2009). Wei et al. clearly showed that the* six3 *gene identified from the California purple sea urchin is first expressed in the animal pole domain at the early blastula stage. The expression in the animal pole can be observed throughout gastrulation in addition to a second expression domain “in some secondary mesenchyme cells scattered throughout the blastocoel and at the tip of the archenteron.”*

In situ time course was extended to include 6hpf (the time of *six3* early blastula expression) and clarified this in the corresponding figure legend. We repeated in situs in later embryos with a new, freshly made probe, and confirm that we see a higher level of expression in the NSM compared to the apical plate. However, we feel our in situs now more closely reflect that of [42] with the inclusion of earlier stages and darker stained embryos from earlier than prism-staged embryos.

*In the subsection “Perturbation analysis unveils a GRN circuit that controls homing”, the authors mentioned “*six3 *was found to be expressed in the aboral mesoderm”, and cited*
[43]*. However, whether* six3 *is expressed in the oral or aboral coelomic pouches or even in the small micromeres requires double staining of six 3 with appropriate markers. Where* six3 *is expressed is important for interpretation of the data. In*
Figure 5*, it is intriguing that knockdown of* six3 *in both micromeres and the host affected homing. Therefore, it is possible that* six3 *is expressed in both small micromeres and the coelomic pouch mesodermal cells.*

These experiments were done and can be seen in Figure 6. Double in situ hybridization was done with *six3* and *eya* to show co-localization with the previously published aboral expression domain (Luo, 2012). We find that *six3* expression does overlap with the aboral expression of *eya,* but also exhibits expression in more oral regions of the coelomic pouch. We also did DFISH with *vasa,* a small micromere marker (Figure 6). It was seen that *six3* does not overlap in expression with the small micromeres, but the cells are immediately apposed.

*One additional issue is that Luo and Su (PLoS Biology, 2012) showed that in addition to the three factors studied by the authors (*dach1*,* eya *and* pax6*),* six1/2 *is also expressed in the aboral coelomic pouch. I wonder whether the authors have investigated this obvious and putative player of the conserved RDGN network.*

The authors would like to thank the reviewers for this suggestion. Six1/2 was added to the story: knockdown surgeries (Figure 5), expression pattern (Figure 6—figure supplement 1), incorporation into GRN (Figure 7), and morphology of morpholino (Figure 7—figure supplement 1). We hope that the addition of Six1/2 further completes the story and also solidifies our claims of this network being incredibly conserved among animals and processes.

8) It is not clear the labeled red cells are migrating to their final site, or are drawn by static cells. This author showed some time ago that the embryo has very long processes that extend across the blastocoel even. Better documentation of the movement of the red cells is needed. What is present appears like a straight line – as might be suggested by a retracting process from a static cell instead of migration.

True – it is possible that the NSM extend filopodia that retract and draw the small micromeres to the coelomic pouches, and we agree that this is a good point. Campanale et al. has published that the small micromeres extend filopodia, suggesting that they are searching for either a signal (chemotaxis) or other filopodia from the coelomic pouch NSM. From our time-lapse data on ectopic ingression (not shown), we see the ectopic small micromeres migrate in a directed fashion to the mesenchyme blastula vegetal plate where NSM is specified. From this data, we hypothesized that the likely mechanism is chemotaxis. However, to test this hypothesis would be out of the scope of this paper; it would require being able to selectively, and permanently remove all of the filopodia of the NSM or small micromere would be infeasible at this time. We added discussion points as to what other mechanisms might be at play.

The authors claim that the small blastomeres are not germ cells, or primordial germ cells, because the lineage of the line has not been documented. They conclude that the cells are multipotent but fail to provide such evidence needed for this conclusion.

We did not refute that the small micromeres are PGCs, and based on the work published in the Wessel lab, we believe that they normally do contribute to the PGC lineage. However, the possibility remains that small micromeres are a multipotent population that not only contributes to the PGCs but also give rise to other adult tissues. This is still an unanswered question, so we agree that calling them “multipotent progenitors” may seem unwarranted. We have changed “multipotent progenitors” to small micromeres in the few instances it was mentioned. Whether or not the small micromeres are multipotent is tangential to the story, so we decided to just remove the wording.

9) Do any cells die? Better quantitation is necessary and they need to document cell death.

Assuming the question of any cells die is meant to be for transplant efficacy, the percentage of embryos with failed transplants was documented along with the original scored surgeries. A “failed transplant” included transplants that did not incorporate into the host embryo at all (no green fluorescence present at 2dpf), transplants where the small micromere did not incorporate but the large micromere did (only skeletal elements were observed), and transplants that were obviously dead within the blastocoel (large, misshapen cells surrounded by cell debris). We listed average percentages of successful and failed transplants for each of the microsurgeries in a table we titled “Transplant Efficacy” (Table 1).

If reviewers are asking for the number of endogenous small micromeres that die, this was not scored.

10) How can the authors distinguish if the MO is specifically testing homing, versus aspects of earlier development?

The transcription factors tested were specifically picked for their later expression in the NSM just prior to the homing ability, so that early specification events would not be perturbed.

11) The methods for statistical analysis need to be described.

Details of the statistical analysis were added to the Materials and methods section. GraphPad software was used to analyze a 2x2 contingency table with groups corresponding to the condition (controls and knockdown micromeres or controls and knockdown hosts) and outcomes being “homing” or “no homing”. Chi-squared was then calculated, and a p-value was significant at a value of less than 0.05.

12) Issues with figures:

Figure 3—figure supplement 1
*is confusing. The ectopic cell is labeled in red in 1A. However, it is not clear whether the red cells migrated into the coelomic pouches. An indication of coelomic pouches or counter stained the embryos with DAPI would help to see the location of coelomic pouches. Similar problem in B. The ectopic cell is now in green. I see there are two cells with green membrane and red nuclei. Are they ectopic or endogenous cells?*

This figure was redone to include DAPI staining, and the ectopic cells are both green and the endogenously placed micromeres are both red. The authors hope that this clarifies any confusion of the figure.

Figure 2*, it is not clear whether a micromere or a small micromere was transplanted? If a micromere is transplanted, how can the authors be sure the two cells breached the laminin layer (indicated with a yellow arrow) at 10 hpf are small micromeres but not large micromeres?*

For this experiment, only small micromeres were transplanted. The authors agree that if a micromere were transplanted, it would be unclear what was causing the breach in the laminin layer. This was clarified in the text (subsection “Small micromeres “home” to the coelomic pouches from ectopic positions”, last paragraph).

The authors provided data to validate unpublished morpholinos using a second morpholino against Eya and Pax6 in Figure 6–figure supplement 2. Please provide references if Six3 and Dach1 morpholinos used in this study have been published and validated before.

Six3 and Dach1 MO were both previously published and validated (Wei, 2009; Peter, 2011). This is now referenced in Table 3.

*Clear development delay is seen in the morpholino-injected embryos presented in*
Figure 6*. This may cause problem especially in the embryos stained with the* six3 *probe. Looking at the* six3 *expression pattern in*
Figure 6—figure supplement 1*4,* six3 *transcript is barely detected. At the similar stage (based on their similar morphology) showed in*
Figure 6*, no expression of* six3 *is interpreted as downregulation. Please clarify the morpholino effect on developmental delay and their effects on* six3 *expression at a comparable developmental stage.*

Good point. All embryos were fertilized, injected, and subsequently fixed at the same times. We do agree that it appears the morpholino injections for Dach1, Eya, and Six3 appear delayed, but we attribute this to their arm phenotype (observed in Figure 7—figure supplement 1).

*Other issues concerning*
Figure 6*: (1)* six3 *expression expands in* six3 *morphants, but the expansion is mostly seen in the apical plate. Therefore, the self-repression is in the apical plate not the coelomic pouches; (2)* six3 *expression seems upregulated in the hindgut of the* eya *morphants. Please confirm whether it is a consistent result.*

(1) *six3* expression expanded in the apical plate only is addressed in revision 1. We agree that the data does not show overexpression in coelomic pouches and have taken the linkage out of our circuit diagram; (2) *six3* expression in the hindgut of Eya morphants was an artifact and after revisiting our slides, we were able to replace the picture with background staining in the gut with a more representative photo (Figure 7).

Figure 3
*presented percentages of the embryos showing small micromere homing. During normal embryogenesis, small micromeres migrate to the left or right coelomic pouches in a 5:3 ratio. Is there any left-right preference of the small micromere homing when they are transplanted at various positions?*

When scoring transplants, we have seen the small micromeres separate into the two coelomic pouches, however, this is rarely seen, and, interestingly, they almost always both end up in the left coelomic pouch. A sentence discussing this was added to the text (“Interestingly, the majority of transplants were scored to be in the left coelomic pouch after homing suggesting a survival mechanism to ensure the proposed primordial germ cells make it onto adulthood to contribute to the gamete pool”).

Figure legend 2D: please indicate what the white and yellow lines are.

White lines indicate the endogenous site of laminin remodeling and PMC ingression. Yellow lines indicate the ectopic breach in laminin by the ectopic small micromeres. This was clarified in the figure legend.

Figure legend 3: (B) should be equator and (C) animal pole.

Thank you. This was corrected.

[Editors' note: further revisions were requested prior to acceptance, as described below.]

[…] The manuscript has been improved but there are some remaining relatively minor issues that need to be addressed before acceptance, as outlined below. Please address the points raised by reviewers 2 and 3 in a revised manuscript:

Reviewer #2:

The authors have addressed most of my concerns and have added substantially to the manuscript. I believe the paper should be accepted for publication. I do have one small quibble: just because a MO was published previously does not mean that appropriate controls for specificity and efficacy were carried out. If so, then the authors point is well taken. If not, then the fact that someone was able to publish an experiment with the MO is irrelevant.

Citations referenced in Table 3 indicated the article in which the published morpholinos had been previously validated. To further emphasize this point, we expanded upon this in the Materials and methods section.

Reviewer #3:

*The authors have answered most of my questions. One remaining issue is the exact expression domain of* six3 *in the coelomic pouches. The vasa signal in*
Figure 6
*is extremely weak. It is difficult to judge whether there is an overlapping domain between* six3 *expression domain and* vasa *localization. The authors concluded that there is no overlapping and* six3 *is not expressed in the small micromeres. If this is the case, the authors need to explain the defect in homing when* six3 *was knockdowned in the small micromeres.*

In order to clearly show the lack of co-localization of *six3* and *vasa*, we added insets to figure panels (Figure 6) at a higher magnification and without the DAPI channel (which we feel may have added to the apparent weakening of the original vasa signal). We feel it is now clear that the two territories are immediately apposed but remain distinct in location.

To further address whether Six3 has a homing role in the small micromere, we focused efforts on creating a larger “n” for “Six3-MO micromere” surgeries scored, which was relatively low prior to these additions (Figure 5). We found that after increasing the number of scored cases, the effect of Six3 when knocked down in just the small micromere was statistically insignificant. From these data, we concluded that Six3 only has a role in homing in the NSM.